# WHO Workshop Report: Regulatory Science to Inform Clinical Pathways for Shigella Vaccines Intended for Use in Children in Low- and Middle-Income Countries

**DOI:** 10.3390/vaccines13050439

**Published:** 2025-04-23

**Authors:** Robert W. Kaminski, Patricia B. Pavlinac, James A. Platts-Mills, Elizabeth T. Rogawski McQuade, William P. Hausdorff, Richard A. Isbrucker, Kirsten S. Vannice, Marco Cavaleri, Sonali Kochhar, Kirsty Mehring-LeDoare, Godwin Enwere, Annelies Wilder-Smith, Karen L. Kotloff, Samba Sow, Birgitte K. Giersing

**Affiliations:** 1Department of Immunization, Vaccines and Biologicals, World Health Organization, 1211 Geneva, Switzerland; mehringlek@who.int (K.M.-L.); wildersmitha@who.int (A.W.-S.); giersingb@who.int (B.K.G.); 2Department of Global Health, University of Washington, Seattle, WA 98105, USA; ppav@uw.edu (P.B.P.); sonalikochhar@yahoo.co.in (S.K.); 3Department of Medicine, Infectious Diseases and International Health, University of Virginia, Charlottesville, VA 22908, USA; jp5t@uvahealth.org; 4Department of Epidemiology, Emory University, Atlanta, GA 30322, USA; erogaws@emory.edu; 5Center for Vaccine Innovation and Access, PATH, Washington, DC 20001, USA; whausdorff@path.org; 6Faculty of Medicine, Université Libre de Bruxelles, 1050 Brussels, Belgium; 7Health Canada, Ottawa, ON K1A 0K9, Canada; richard.isbrucker@hc-sc.gc.ca; 8Bill & Melinda Gates Foundation, 440 5th Ave. N, Seattle, WA 98109, USA; kirsten.vannice@gatesfoundation.org; 9European Medicines Agency, 1083 HS Amsterdam, The Netherlands; marco.cavaleri@ema.europa.eu; 10Global Healthcare Consulting, Gurgaon 122503, India; 11Vaccine Assessment Team Pre-Qualification, World Health Organization, 1211 Geneva, Switzerland; enwereg@who.int; 12Center for Vaccine Development and Global Health, University of Maryland, Baltimore, MD 21201, USA; kkotloff@som.umaryland.edu (K.L.K.); ssow@cvd-mali.org (S.S.); 13Centre pour le Développement des Vaccins, Bamako P.O. Box 251, Mali

**Keywords:** *Shigella*, vaccines, children, phase III study, endpoints, safety, low- and middle-income countries

## Abstract

Infectious diarrhea caused by *Shigella* remains a significant global health concern, and several vaccine candidates are approaching phase III clinical studies in the target population of young children in low- and middle-income countries. The World Health Organization (WHO) has published preferred product characteristics (PPCs) for *Shigella* vaccines to provide strategic guidance that aids in advancing product development and highlights policy considerations for use in LMIC settings where the vaccine is most needed. However, the selection of appropriate clinical endpoints was not clearly defined within the PPCs and remains an important issue for phase III study design. Previously, an expert panel identified areas of alignment and consensus on many clinical study design components while also recognizing that further discussions and data were required to solidify recommendations on key study design aspects. Therefore, WHO convened a diverse range of stakeholders, including manufacturers, regulators, and policymakers across national, regional, and global levels, with the aim of achieving consensus and soliciting inputs from the regulatory community surrounding vaccine phase III study design considerations. The intent of this report is to outline the key points from those discussions to inform the phase III design strategies and investment decisions of product developers and donors and to share recommendations for next steps.

## 1. Introduction/Background

Diarrheal disease remains a significant global health threat, with an estimated 1.1 million diarrheal-related deaths in 2021, affecting all age groups across diverse geographical regions [1,2]. However, the most significant impact is among children under 5 years old in low- and middle-income countries (LMICs). *Shigella* remains a leading cause of diarrheal disease worldwide leading to an estimated 94,000 deaths in 2019, with the greatest burden of morbidity and mortality seen in children between 6 months and two years of age [3]. Both symptomatic and asymptomatic *Shigella* infections in young children can also lead to growth faltering and stunting [4,5], with long-lasting detrimental effects on physical and cognitive development [6], increasing susceptibility to other infectious diseases. Thus, preventing *Shigella* infections through improved sanitation, vaccination, and ready access to healthcare and antibiotic treatment is essential for reducing the risk of stunting, promoting healthy growth in children, and allowing populations to reach their fullest economic and productivity levels. However, the overuse and misuse of antibiotics contribute to the emergence and spread of antibiotic-resistant strains of *Shigella*, posing a serious public health threat [7,8] recognized by WHO and the US Centers for Disease Control. The WHO Global Antimicrobial Resistance Surveillance System [9] identified *Shigella* as a priority pathogen for the development of new interventions. This underscores the importance of addressing antimicrobial resistance (AMR) through appropriate antibiotic stewardship practices, improved sanitation and hygiene measures, and the development of effective vaccines against *Shigella*.

Several multi-valent *Shigella* vaccines are currently in development [10] designed to target *S. sonnei*, *S. flexneri* 2a, 3a, and *S. flexneri* 6 or 1b, with phase II studies in infant target populations underway or recently completed in Africa and Asia. To further support vaccine introduction, WHO has prioritized defining pathways and evidence requirements for developing safe, effective, and affordable vaccines to combat *Shigella*-induced dysentery and diarrhea in children under 5 years old in LMICs. This strategy aims not only to expedite regulatory approval for the target group but also to ensure the availability of evidence to support efficient policy recommendations and timely implementation. Consequently, WHO has developed preferred product characteristics (PPCs) for *Shigella* vaccines [11], outlining the desirable product attributes for their utilization in LMIC settings and contributing to better health outcomes in vulnerable populations. The WHO PPCs emphasize the need for vaccines that are safe, affordable, and provide long-lasting protection, particularly for infants and young children in LMICs. Key product characteristics outlined in the PPCs include an acceptable safety and reactogenicity profile, efficacy in preventing moderate to severe *Shigella* diarrhea in children from 6–36 months, ease of administration, and stability in varying storage conditions. The PPCs also stress the importance of vaccines that can be integrated into existing immunization programs to maximize coverage and impact.

The clinical pathway to marketing authorization for *Shigella* vaccine developers and funders has been a critical question during the early phases of the product development [12]. A previous WHO-led consultation of experts, policymakers, and regulators determined that an efficacy trial would be on the critical path for licensure for infants in LMICs given the feasibility of the trial, the lack of an established correlate of protection, and the concern of generalizability of the adult controlled human infection model (CHIM) to children living in LMICs [13]. A phase III study of a bivalent (*S. flexneri* 2a and *S. sonnei*) conjugate vaccine is underway (NCT05156528), with other trials likely to follow in the coming years. Additionally, a large multi-country observational study is underway to identify potential trial sites and characterize baseline incidence [14,15,16]. The Enterics for Global Health (EFGH) *Shigella* surveillance study [16,17] employs cross-sectional and longitudinal study designs to establish incidence and consequences of medically attended *Shigella* diarrhea in children 6–35 months of age in Bangladesh [14], Gambia [18], Kenya [19], Malawi [20], Mali [21], Pakistan [22], and Peru [15].

The selection of appropriate clinical and microbiologic parameters for a phase III study remains an important issue since they can significantly impact study power and efficacy estimates due to the relative sensitivity and specificity of the options for microbiologic confirmation and because the vaccine may preferentially prevent a more severe phenotype rather than all disease. Therefore, an ad hoc expert panel was convened to outline phase III study design considerations for a *Shigella* clinical study in infants from LMICs [23]. The panel identified areas of alignment and consensus on many aspects of study design consideration for a phase III vaccine trial while recognizing that further discussions, data, and regulatory input were needed to solidify recommendations on key aspects. To achieve this goal, WHO convened a diverse range of stakeholders, including vaccine manufacturers, regulators, researchers, and policymakers across national, regional, and global levels, to facilitate discussions aimed at building consensus on *Shigella* vaccine phase III study design considerations.

## 2. WHO Regulatory Science Workshop

The Regulatory Science Workshop held in Nairobi, Kenya, in 2024 was designed around four discussion domains (see Table 1): (1) vaccine safety considerations, (2) primary clinical endpoints including case definitions and case ascertainment methods, (3) secondary clinical endpoints, and (4) exploratory or descriptive endpoints that could positively impact policy-making decisions. The two-day participatory and interactive workshop had the overarching goal of gaining consensus and informing potential clinical endpoints, case definitions, and case ascertainment methods to better navigate regulatory approval and policy framework. Over 35 regulators from the AFRO, AMRO, EMRO, EURO, and SEARO WHO regions participated in the workshop, particularly those from countries in which the phase III field efficacy study may be conducted. Input from the regulatory community was sought through a series of round table, guided discussions to align clinical endpoints, clinical case definition, and case ascertainment method(s) with regulatory expectations. The intent of this report is to outline the key points from those discussions and further inform the phase III design strategies and investment decisions of product developers and donors.

## 3. Vaccine Safety Considerations

Several *Shigella* vaccines have progressed through Phase I and Phase II studies in adult, child, and infant populations, with vaccine safety being a key endpoint of those evaluations. The safety endpoints being assessed in those Phase IIa *Shigella* vaccine studies (NCT04056117, NCT04602975, NCT05073003) include solicited adverse events (AEs) for 7 days, unsolicited AEs for 28 days, and serious AEs (SAEs) and adverse events of special interest (AESIs) for the trial’s duration.

There was consensus among the workshop participants that the safety data collected from the previous Phase IIa studies were also appropriate for Phase III studies and should continue to be reported as per standard regulatory guidelines by relatedness, grade, and frequency, with Data and Safety Monitoring Boards (DSMBs) in place for a thorough oversight. Regulatory guidelines, such as those from the FDA and EMA, typically require assessing vaccine safety in a minimum of 3000 study participants. A robust follow-up and monitoring system is crucial to ensure adequate reporting of AESIs. In the context of a Phase III *Shigella* vaccine study, unexpected AEs and immune-mediated diseases should be captured and reported. Additionally, post-marketing commitments such as pharmacovigilance/surveillance and a Risk Management Plan may be required, along with consideration of regional factors. No additional safety endpoints were specifically outlined, but long-term follow-up if co-administration with other vaccines occurs and investigations into comorbidities, such as HIV-exposed subjects and HLA-related autoimmunity, may be anticipated.

## 4. Primary Vaccine Efficacy Endpoint

There was broad consensus that the primary endpoint for a phase III *Shigella* vaccine study in infant and toddler populations from LMICs should demonstrate vaccine efficacy against only those *Shigella* serotypes targeted by the vaccine (e.g., *S. flexneri* 2a, 3a, and 6 and *S. sonnei*). Furthermore, it was considered important that efficacy should be associated with the first episode of *Shigella*—attributed diarrhea in the trial—as repeat Shigella exposures may be a confounder. It was recommended that a phase III primary endpoint should be vaccine efficacy against the first episode of moderate or severe diarrhea caused by *Shigella* serotypes targeted by the vaccine, aligned with the WHO PPCs and past phase III trials of rotavirus vaccines, to demonstrate a strong public health value. Discussions also identified topics related to the primary endpoint that required additional alignment with regulatory expectations, including clinical case definition(s) and case detection methodologies.

### 4.1. Clinical Case Definition

Prior to the meeting, there was agreement that more severe diarrhea (as opposed to all diarrhea) would be required as the primary clinical case definition based on the PPCs. Therefore, several clinical case definitions of moderate-or-severe disease (MSD) were discussed with regulators to identify the most suitable definition to serve as a framework for a Shigella Phase III vaccine study. A dichotomous definition [24] was developed for the Global Enteric Multicenter Study (GEMS). The GEMS-MSD definition utilizes diarrhea (≥three abnormally loose stools in 24 h) accompanied by dehydration and/or dysentery (at least one stool containing visible blood according to either the caretaker or the clinician) as criteria.

The modified Vesikari (mVesikari) scoring system, which has been utilized in several enteric vaccine studies, including rotavirus [25], assigns a score across a wide range of signs and symptoms, including duration and maximum number of stools, maximum number and duration of vomiting, temperature, dehydration, and treatment. The scores have been used to rank diarrhea severity into mild (0–6 points), moderate (7–10 points), or severe (≥11 points) illness categories.

Several key attributes of the mVesikari and GEMS-MSD definitions were discussed to ensure the ability to accurately and robustly identify cases that aligned with the trial clinical endpoints of measuring vaccine impact on MSD. The mVesikari score emerged as a potential case definition, yet concerns were raised regarding its applicability to *Shigella*, notably the emphasis of the scoring system on vomiting, which is not as predominant with *Shigella* infections as with rotavirus. It was also noted that the severity definition does not need to distinguish *Shigella* from other etiologies since etiology in a phase III Shigella vaccine study will be determined through culture and/or qPCR methods. Additionally, the score’s interpretation and complexity were noted. As an alternative, a refined case definition based on GEMS-MSD criteria was proposed, aiming for simplicity and relevance to *Shigella*, with some experts suggesting an amended GEMS-MSD definition should include diarrhea duration and fever. The discussion also touched upon the importance of clinical case definitions in the context of secondary endpoints and post hoc analyses, particularly in evaluating the vaccine’s impact across multiple disease severity subsets. Therefore, a recommendation from the workshop included establishing an expert working group to optimize a case definition modeled after the mVesikari and GEMS-MSD definitions, which would include an objective, simple composite case definition that easily explains the clinical syndrome of more severe shigellosis while also adding additional details than are currently captured in the GEMS-MSD definition. Importantly, the case definition should also be able to disaggregate moderate from severe cases and be more fully aligned with characteristics of *Shigella* disease. The recommended case definition would then be reviewed during a future WHO expert consultation for broader input and alignment prior to generating a meeting report to disseminate finalized recommendations to the global community.

### 4.2. Microbiological Confirmation Methodology

Classical microbiological culture methods and quantitative polymerase chain reaction (qPCR) are two prevalent techniques for detecting *Shigella* in stool samples and have been utilized in numerous epidemiological studies [26,27]. Both have distinct advantages and disadvantages in terms of sensitivity, specificity, time efficiency, and practical application in clinical settings. As the detection methods have implications for trial size, cost, and feasibility, as well as the ability to compare trial results, it would thus be ideal to formulate a clear recommendation as part of the case definition.

One significant advantage of qPCR is high sensitivity. qPCR can detect low levels of *Shigella* DNA, making it highly effective, even in cases with low bacterial loads or after antibiotic treatment that may not be detected by culture methods [28,29]. Additionally, qPCR allows for (1) rapid processing and results, often within a few hours, which is crucial for timely diagnosis and treatment in clinical settings, or alternatively (2) sample batching to allow simultaneous analysis across multiple samples or centralizing the qPCR analysis during a multisite study, which can positively impact reproducibility. *Shigella* detection by qPCR is accomplished by amplifying the *ipaH* gene, which is also found in enteroinvasive *E. coli* (EIEC). Moreover, qPCR can sometimes detect non-viable bacteria, leading to lower clinical specificity since it amplifies DNA regardless of the bacteria’s viability. Recent advances, such as setting cycle threshold cut-offs that optimize specificity, as well as adding serotype specific targets for *Shigella* may overcome these concerns. Also, combining qPCR with a clinical case definition that captures more severe disease will decrease likelihood of asymptomatic detection of *Shigella*.

Culture methods have the advantage of confirming the presence of viable *Shigella* organisms and allow for antibiotic susceptibility testing, which is crucial for selecting appropriate treatments and informing potential exploratory clinical study endpoints. Culturing is generally more cost-effective compared to qPCR and does not require sophisticated equipment. technical training. Despite these benefits, culture methods have notable disadvantages, including being time-consuming, typically requiring 24–48 h or longer to yield results, which delays diagnosis and treatment, and requiring extensive training. Additionally, culture methods are difficult to standardize across laboratories and have lower sensitivity compared to qPCR [28], particularly if the bacterial load in the sample is low or if the bacteria are in a viable but non-culturable state due to prior antibiotic use or sample handling conditions, as *Shigella* can be sensitive to temperature, pH, and oxygen levels during transport. This can result in false-negative results and potentially undetected *Shigella* cases, especially among younger children who are less likely to be culture positive. Moreover, the lower sensitivity substantially impacts clinical trial sample size calculations and therefore may affect the feasibility of successfully conducting a phase III study.

There is an inherent relationship between microbiological methods for confirmation of cases and sample size required to power a phase III pediatric efficacy trial to detect the target vaccine efficacy of at least 60% against vaccine-preventable *Shigella* strains, as outlined in the WHO PPCs. For example, using qPCR for confirmation of *Shigella* infection increases the trial’s power compared to traditional culture methods. The calculations show that for a primary endpoint of mVesikari ≥9 and/or dysentery with qPCR confirmation for vaccine-preventable strains, a one-year follow-up trial could require ~6000 participants, based on incidence estimated at some EFGH sites. In contrast, using less sensitive culture confirmation would require >15,000 participants to sufficiently power the study. Therefore, qPCR for *Shigella* detection has emerged as an attractive alternative to classic culture methods. Furthermore, advancements and refinements over the past several years have also facilitated the use of qPCR as a means of determining *Shigella* species and serotype directly from stool samples [30,31], further increasing the practicality of the assay with the current PCR serotyping methodology capable of detecting vaccine-preventable serotypes proposed in a phase III study, which include *S. sonnei* and *S. flexneri* 2a, 3a, 6, and 1b [31].

When discussed with the regulatory community, it was noted that while some regulators may prefer culture-based methods due to their ability to definitively detect *Shigella*-attributed diarrhea, regulators are open to considering PCR if appropriate data are generated to demonstrate that it is fit-for-purpose and possible to validate, which could support regulatory acceptance. It was emphasized that clarity on the diagnostic’s intended use and interpretation of discordant results between culture and PCR is crucial, with pre-specification in protocols and analysis plans recommended. In particular, justification of the PCR cycle threshold (ct) for selection of actual cases needs to be discussed with regulators before starting the clinical trial. There was a suggestion that the ICH Q2(R2) guideline, which focuses on standard analytical metrics (sensitivity/specificity, range, linearity, limit of detection, accuracy (via spiking known quantities of analyte into a test sample), precision (via reproducibility), and reproducibility (between labs)), is a good resource for understanding data required for licensure submission [32,33]. Much of these data are available or published for the TaqMan Array Card [34,35].

The importance of engaging with regulatory bodies to design the appropriate data package early in the process was underscored. For example, the data package would be different if the qPCR assay was being used as a clinical diagnostic test versus as a biomarker to support clinical endpoints, with the latter not typically requiring formal licensure. It was suggested that a “Type C meeting” with the FDA could be one mechanism to provide guidance once clarity on the intended use and limitations of the assay were established and that engagement with the FDA or EMA, in collaboration with LMIC regulators from countries in which the vaccine is intended to be used, could help pave the way for rapid regulatory approval in LMICs.

Other points of discussion indicated that derivation and validation of a quantitative cutoff could possibly face more scrutiny, and data supporting the clinical specificity of *Shigella* detection at that cutoff would be crucial. Similarly, data would be needed to support the use of a quantitative algorithm to serotype *Shigella*-attributable diarrhea using O-antigen synthesis and modification genes. A semi-quantitative assay would be acceptable, but converting cycle thresholds to copy numbers was advised. Finally, a centralized lab could be used for a multisite study, but if not, data will have to be presented demonstrating the consistency of test performance between sites. Additionally, lessons could be drawn from the use of PCR in malaria vaccine trials [36,37]. Overall, the discussion underscored the importance of robust data, clear regulatory engagement, and contextual clarity in utilizing PCR for *Shigella* detection. Furthermore, regardless of which laboratory method is used for the primary endpoint and determining sample size, both qPCR and culture methods should be employed.

## 5. Secondary Endpoints

For a phase III Shigella vaccine clinical trial, primary endpoints will focus on key measures to determine vaccine efficacy, such as the prevention of laboratory-confirmed Shigella infections or a significant reduction in severe gastrointestinal symptoms among vaccinated individuals compared to a control group. These endpoints are the trial’s main outcomes, guiding regulatory and clinical approval decisions. Secondary endpoints explore additional aspects of vaccine performance, like reduction in all-cause diarrheal disease, duration of illness, or immune response levels, offering further insights into the vaccine’s broader effects. Exploratory endpoints investigate less established or long-term outcomes, such as identifying biomarkers of immunity, evaluating vaccine impact across different age groups or geographic regions, or monitoring potential off-target benefits. These endpoints are primarily hypothesis-generating, potentially guiding future vaccine development and public health strategies.

Potential secondary clinical endpoints were also discussed, priority ranked, and, in some cases, modifications were suggested. The highest priority endpoints were largely focused on (i) vaccine efficacy against a broad range of disease severities, including less-severe disease, followed by (ii) vaccine efficacy irrespective of *Shigella* serotype, and (iii) evaluating the longevity of the protective immune response induced through vaccination (Table 2). A secondary endpoint that provides flexibility surrounding which clinical syndrome is vaccine-preventable was an important attribute considered. Using a case definition scoring system, such as mVesikari and/or dysentery, allows us to identify protection against a variety of severity subsets to be evaluated as secondary endpoints in a transparent and feasible manner, whereas there was less clarity on how the GEMS-MSD definition could be categorized into severity subsets. If a new, non-scoring system is developed, it was agreed that it should at least separate mild, moderate, and severe to provide dimensions for endpoint disaggregation. Additionally, if a phase III Shigella vaccine study were to enroll a wide age range (e.g., 6–60 months), establishing efficacy by age cohort would be an important aspect for consideration as a secondary endpoint since these data would be critical to understanding the impact of different vaccination schedule options and have important implications surrounding the role of pre-existing immunity or boosting through pathogen exposure in endemic settings.

The second highest priority was to evaluate vaccine efficacy against more *Shigella* serotypes than those specifically targeted within the vaccine or to assess vaccine-induced cross-protection. There was a recognition during the discussion of challenges that may be presented depending on the choice of case detection methodologies, specifically culture versus PCR, largely based upon the limitations of qPCR to distinguish beyond *S. sonnei* and *S. flexneri* serotypes. However, there were strategies discussed that could partially mitigate these challenges, such as limiting the secondary endpoint to all *Shigella flexneri* serotypes and *S. sonnei* if PCR was utilized as the case detection method employed in the phase III study. Alternatively, both culture and PCR-based detection methods could be employed if protection against less prominent Shigella serotypes (e.g., *S. boydii*) were to be assessed.

The third priority tier for secondary endpoints included measures of vaccine efficacy duration by capturing not only the first episode of *Shigella* diarrhea after vaccination but throughout the course of the study, which was shown to be valuable to policy decisions in the malaria vaccine field [38]. However, it may be difficult to substantially power the study and may require alternative statistical methods, such as survival analysis, to effectively capture vaccine impact. Several other clinical endpoints were also suggested for consideration, including determining vaccine efficacy (1) stratified by follow-up time, (2) against all-cause diarrhea meeting primary case definition, (3) against caregiver-reported diarrhea, including episodes that did not lead to care-seeking, and (4) stratified by baseline antibody levels. Two additional suggestions were to include subgroup analyses considering country/site, sex, and age and, if budget allows, to evaluate inflammatory markers [39] for vaccine breakthroughs versus unvaccinated cases.

## 6. Exploratory Endpoints

As the discussions turned to exploratory endpoints, an emphasis was placed on distinguishing between hypothesis-driven and descriptive endpoints. Hypothesized (regulatory) endpoints are pre-specified primary and secondary outcomes used to assess the efficacy and safety of an intervention. The primary endpoints are statistically powered and critical for gaining regulatory approval from bodies like the FDA or EMA. Conversely, descriptive endpoints are exploratory outcomes that provide broader insights into the intervention’s real-world impact that may be informative for policy decisions, including long-term safety and effectiveness. All types of endpoints are essential for a comprehensive evaluation of the vaccine’s benefits and implications. In the context of a phase III *Shigella* vaccine study, exploratory endpoints associated with the vaccine’s impact on AMR/antibiotic usage and growth faltering/stunting were discussed by the convening.

With respect to vaccine impact on growth faltering, the regulatory community asserted that assessing growth is feasible and utilized in nutritional studies, although there was speculation about its relevance at this stage of clinical vaccine development. Concerns were expressed regarding the interpretation of individual z-scores, misclassification of infection, and impact of environmental enteric dysfunction, collectively affecting growth impact. Furthermore, treatment with antibiotics, which would likely be included as standard of care in a phase III Shigella vaccine trial, has been shown to ameliorate impacts of infection on growth [4]. While not crucial for regulatory approval, there was broad agreement that demonstration of vaccine impact on growth faltering would likely be well-received by policymakers and facilitate vaccine introduction decisions. However, large-scale anthropometric data collection requires significant resources, and the expected effects of the vaccine on growth are likely to be very small (<0.1 z-scores for individuals with shigella and <0.01 z-scores on average for the whole trial population). Attempting to observe vaccine effects on overall growth was compared to observing vaccine impacts on all-cause mortality, which is typically beyond study power. Underpowered studies pose high risks; a lack of demonstrated growth effects could dampen vaccine enthusiasm, and caution was expressed to avoid negative results and opportunity costs. Therefore, there was the suggestion that endpoints related to growth faltering may better align with a descriptive exploratory endpoint at this stage of clinical development. Such descriptive endpoints could include changes in anthropometric measures by vaccination status, for example. Length for age (LAZ) at regular intervals (6, 12, 18, and 24 months) in the trial or other pragmatic times based on the regularly scheduled visits can be collected in the trial, but there remains a likelihood post-licensure study data would be needed to fully inform these descriptive exploratory endpoints.

The tremendous impact of antimicrobial resistance (AMR) on communities was well-recognized across various stakeholders, highlighting the importance of strategic national control and policy formulation. There was a suggestion that in phase III trials, an opportunity exists to compare antibiotic usage between vaccinated and unvaccinated groups, with routine data collection on concomitant medications during both scheduled and unscheduled visits. Health books or diaries could be used to cover gaps between visits, supported by additional phone calls or decision support systems (DDS) for comprehensive data collection. Therefore, including a descriptive exploratory endpoint to document the impact of a *Shigella* vaccine on antibiotic usage was recommended even if it would not be included in the vaccine indication since it would be valuable data for policy decisions. It was suggested that a statistically significant difference in antibiotic use between vaccine and placebo groups could positively influence policymakers, especially in LMICs, by demonstrating potential cost and budget impacts, as well as contributing to global AMR reduction efforts. It was noted that training healthcare providers on the vaccine’s impact on AMR is also beneficial. Interestingly, there was consensus that if the additional benefit of reduced antibiotic usage was not substantially demonstrated, it would signify the absence of an extra benefit rather than posing a definitive risk. It was also noted as a relevant consideration that if the phase III study was powered for a primary outcome based on PCR, AMR endpoints would be underpowered because they can only be assessed in culture-positive cases. Therefore, meeting participants recommended convening a WHO Policy Workshop with LMIC stakeholders to further discuss how descriptive exploratory endpoints such as reductions in stunting/growth faltering and AMR/antimicrobial usage from a phase III *Shigella* vaccine study can support policy decisions, vaccine implementation, and uptake in LMICs.

## 7. Correlate Generation and Facilitation of Clinical Studies

In addition to clinical objectives and endpoints, there were also discussions at the workshop surrounding opportunities for correlates of protection (CoP) to be generated as part of the phase III trial. CoPs may empower immunobridging from demonstrated vaccine effectiveness in a phase III efficacy trial conducted under one set of conditions to infer vaccine effectiveness under another set of conditions, such as in a different age group or demographic group, concomitant administration with other vaccines, or with different Shigella vaccine platforms. Immunobridging could be facilitated in a phase III study through nested immunology cohorts (500–1000 subjects). Several efforts underway to support immunobridging were discussed, including the harmonization of antibody binding assays (LPS-ELISA), functional assays (serum bactericidal assays), and the establishment of a WHO International Reference Serum to support phase III vaccine evaluations.

The workshop convening also discussed opportunities to facilitate clinical studies. For Phase III studies conducted at multiple sites across several LMICs, facilitating joint review by multiple national regulatory authorities or ethics committees, especially in the intended countries of use, can foster consensus on aspects such as optimal case definitions and endpoints, ethical and regulatory matters, and common data requirements. The African Vaccine Regulatory Forum (AVAREF) is an example, aiming to harmonize regulatory and ethics practices across the African continent to support product development. Under the African Medicines Regulatory Harmonization (AMRH) program, AVAREF serves as the technical committee for clinical trials, aligning its tools and processes with the African Medicines Agency (AMA) to support regulatory reliance implementation.

Additionally, the EMA (European Medicines Agency) M4ALL (Medicines for All) program is a strategic initiative aimed at enhancing global access to high-quality medicines. This program focuses on fostering international collaboration and harmonizing regulatory standards to ensure that safe, effective, and affordable medicines are available to all populations, particularly in LMICs. M4ALL provides a mechanism for seeking scientific opinions on high-priority vaccines intended for markets outside the EU, in collaboration with the WHO. This process leverages local epidemiology and disease expertise to provide a unified development and assessment pathway, aiming to expedite WHO prequalification and registration in target countries. By working with various stakeholders, including other regulatory agencies, pharmaceutical companies, and international organizations, M4ALL seeks to streamline the development, approval, and distribution processes for essential medicines, addressing critical public health needs worldwide. Key components of the M4ALL program include capacity-building initiatives, technical support, and the promotion of best practices in regulatory science. The program also emphasizes the importance of innovation and the use of modern technologies to overcome barriers in medicine accessibility. By creating a more efficient and unified regulatory environment, M4ALL aims to reduce the time and cost associated with bringing new medicines to market, ultimately improving health outcomes and reducing disparities in healthcare access across different regions.

## 8. Recommended Efforts to Further Support Clinical, Regulatory, and Policy-Related Strategic Planning

Several workstreams (Table 3) have been identified during the Regulatory Science Workshop to prepare for and inform phase III clinical studies and further support *Shigella* vaccine approval for use in young children living in LMICs. Many of these recommended workstreams could be conducted in parallel to ensure the most expeditious support.

It was recommended that an expert consultation should be convened to establish a viable clinical case definition for use in the phase III primary and secondary endpoints. A clinical case definition that fully addresses moderate and severe *Shigella* infections in young children in an objective and reproducible manner while allowing differentiation of disease severity could facilitate critical secondary endpoints that will support regulatory approval and market authorization while providing critical input needed for policy and vaccine implementation pathways.

Similarly, it was recommended that experts should engage in the qualification and validation of a qPCR methodology to be used for clinical case detection. Data sets from the EFGH *Shigella* surveillance study could be utilized to compare and bridge between the microbiological culture [40] and qPCR methods [34]. These efforts may culminate in a type-C meeting with the US FDA (or equivalent interaction with a regulatory body in LMICs) to facilitate early consultations on the use of the PCR assay to support the primary endpoint in a future Phase III study.

It was also recommended that WHO should convene a policy workshop with LMIC stakeholders, donors, and policymakers to explore how secondary and exploratory endpoints in a phase III study focused on measuring the ability of a *Shigella* vaccine to diminish antibiotic usage and reduce growth faltering/stunting could support vaccine implementation and country uptake. A meeting report from the Policy Workshop should be crafted and published to disseminate outputs, discussions, and recommendations. It is anticipated that the results of the recommended efforts outlined above may necessitate revisions to the WHO preferred product characteristics guidance document, which will be considered and presented to the Product Development for Vaccines Advisory Committee (PDVAC; https://www.who.int/groups/product-development-for-vaccines-advisory-committee (accessed on 20 April 2025)) for external advice and concurrence.

Recognizing that immunization programs around the world are struggling to accommodate the increasing number of vaccines, the logistics of vaccine delivery, financial constraints, and a limited number of medical visits for vaccination, especially in young children, there is an increased urgency to develop combination vaccines [41]. As *Shigella* vaccines enter phase III studies, considerations need to be taken to promote and advocate for vaccine implementation, which may be more successful if a *Shigella* vaccine were combined with another near-licensure or licensed vaccine. Potential *Shigella* combination vaccines have been explored [42] with an emphasis on manufacturing and compatibility, epidemiological and clinical presentations, and strategy, policy, and regulatory considerations. However, there is a need for a strategic framework that can be applied to identify and structure considerations across public health and economic values to better inform and create demand from stakeholders and de-risk investments. Questions regarding the accessibility and finance of approved Shigella vaccines were not considered in the Regulatory Science Workshop, although they can be important as vaccines move closer to market authorization. Although not a workstream directly linked to the Regulatory Science Workshop, efforts are underway to develop the framework needed to evaluate potential combination vaccines, which may help ensure uptake of a *Shigella* vaccine across LMICs with the greatest need.

There is a significant global health burden posed by *Shigella* infections coupled with increasing antibiotic resistance, underscoring the urgent need for an effective vaccine and driving the strategic priority for WHO. However, the demand for a *Shigella* vaccine may not be sufficient, as global awareness of disease burden is lacking, and countries are facing a multitude of competing interventions, including multiple other vaccines, all of which pose huge economic and logistical challenges requiring prioritization. Therefore, significant investment in vaccine demand mobilization through effective communications and community engagement is needed to ensure vaccines become vaccinations.

## Figures and Tables

**Table 1 vaccines-13-00439-t001:** Major discussion domains, complexities, and feedback to inform next steps.

Discussion Domains	Issues/Complexities	Input/Next Steps
Safety Considerations	Are the safety assessments conducted in phase IIa Shigella vaccine studies sufficient for phase III studies, or are there any recommended additions? Can the regulatory community expound on adverse events that would be important to capture in the context of a phase III *Shigella* vaccine study?	Consensus that the safety assessments from phase IIa Shigella vaccine studies are sufficient for phase III studies. Regulatory bodies require assessing vaccine safety in a minimum of 3000 study participants. Unexpected AEs (due to new adjuvant introduction, immune-mediated diseases, and gender-specific issues) should be captured.
Clinical case definitions	Two clinical case definitions were considered for a phase III *Shigella* vaccine study (MSD-GEMS and MSD by mVesikari ± dysentery). MSD-GEMS does not easily account for various disease severities; some parameters are considered subjective. mVesikari ± dysentery allows post hoc analysis of vaccine impact across a range of disease severities; it is not specifically tailored to Shigella disease.	Evaluate the use of MSD-GEMS and mVesikari scoring systems using EFGH data sets. Refine definition(s) to remove subjective measures and ensure robust alignment with Shigella disease. Convene stakeholders, particularly regulators, to review recommended and refined case definitions.
Laboratory confirmation of *Shigella* infection	Culture and qPCR considered for Shigella case detection. Culture allows AST and investigations into vaccine impact on AMR. qPCR is more sensitive, facilitating lower phase III study sample sizes. qPCR capable of serotyping but limited to *S. sonnei* and *S. flexneri* serotypes.	Regulators are open to either culture or PCR as case detection methodology. PCR will require submission of a data package to support phase III usage. Efforts are underway to develop a data package; comparisons in the context of the EFGH study. Both should be performed in a trial regardless of what is used for the primary endpoint.
Secondary endpoints	Primary endpoint focused on vaccine impact on first Shigella infection with a serotype targeted by the vaccine. What would be the recommendation(s) for secondary endpoints in a phase III *Shigella* vaccine study? Provide insights towards modifications or clarifications that would align these endpoints more thoroughly with regulatory expectations. If qPCR was used for case detection, vaccine impact on infection with serotypes other than *S. sonnei* and *S. flexneri* would be limited.	Vaccine efficacy against a broad range of disease severities. Vaccine efficacy across *Shigella* flexneri serotypes not targeted by the vaccine. The duration, longevity, and durability of vaccine efficacy. Vaccine efficacy in the target age group for routine immunization, if a broader age range is evaluated.
Vaccine impact on growth faltering/stunting	*Shigella* is one of many causes of growth-faltering and a trial will not be able to pre-specify Shigella-attributed growth faltering; therefore, it will be underpowered. Strong adherence to effective treatment in a clinical trial may modify vaccine impact on growth faltering. The feasibility to measure growth measurements in infant and young child populations using the WHO Growth Measuring and Standards has been accomplished at LMIC sites in a variety of epidemiology studies (e.g., GEMS, VIDA, ABCD, EFGH). Can the regulatory community comment on the feasibility of collecting sufficient data during a phase III study, over an anticipated follow-up of 2–3 years, to measure vaccine impact on growth faltering?	Nutritional studies have successfully incorporated impact on growth faltering, suggesting feasibility in a Phase III vaccine study. However, study design and sample size may be significantly impacted. Likely more feasible to treat as an exploratory (descriptive) endpoint. Potential to include in post-marketing surveillance and impact studies.
Vaccine impact on antibiotic usage as a proxy for AMR reduction	Measuring reductions in antibiotic usage may be feasible in a Phase III study. Would a reduction of total antibiotic use/burden be sufficient, or does the regulatory community suggest more granular data collection (for example, antibiotic dose utilized, duration of antibiotic usage)? Would a reduction in WHO-recommended antibiotic treatment protocols be sufficient, or should the data sets also focus more widely on classes of antibiotics utilized?	Estimating the impact of a *Shigella* vaccine on antibiotic usage was recommended even if it would not be included in the vaccine indication since it would be valuable data for policy decisions. Consensus that if the additional benefit of reduced antibiotic usage was not substantially demonstrated, it would signify the absence of an extra benefit rather than posing a definitive risk. Convene a WHO Policy Workshop with LMIC stakeholders to further discuss.

**Table 2 vaccines-13-00439-t002:** Recommended prioritization and categorization of secondary endpoints.

Priority	Category	Proposed Secondary Endpoint
1	Vaccine efficacy against broad range of disease severities	Vaccine-preventable subset of *Shigella*-associated disease across a broad range of severities utilizing alternative definitions of disease severity, such as very severe diarrhea and diarrhea of any severity, as determined by mVesikari, as well as dysentery alone.
2	Vaccine efficacy across *Shigella* serotypes not targeted by the vaccine	All *Shigella* moderate or severe diarrhea and/or dysentery episodes, irrespective of serotype
		*Shigella* moderate or severe diarrhea and/or dysentery due to molecularly confirmed *Shigella flexneri* or *Shigella sonnei*
3	The duration, longevity, and durability of vaccine efficacy	Evaluations of protection longevity afforded by vaccination through evaluation of not only first but also subsequent diarrheal episodes
		Efficacy between first and second vaccination (disease between 6 months and 9 months)
No priority	Not recommended	Vaccine-preventable subset of *Shigella*-associated disease using GEMS-MSD and GEMS-LSD definitions
		Vaccine-preventable subset of *Shigella*-associated disease using hospitalization as a severity indicator

**Table 3 vaccines-13-00439-t003:** Recommended future actions to support phase III study design.

**Expert consultation/SME Working Group to Finalize a Case Definition/Scoring System**
Convening of an expert consultation/SME working group to draft a case definition/scoring system for use in a primary endpoint for a *Shigella* phase III study conducted in an LMIC setting. Present the recommended clinical case definition to vaccine developers, donors, and LMIC stakeholders to gain broad input and consensus prior to finalization. Publish recommendations and learnings from the expert panel and stakeholder consultations.
**qPCR Methodology Qualification and Validation to Support Phase III Studies**
Convene an expert group to review existing data sets and identify gaps to be considered to ensure the *Shigella* qPCR assay is “fit for purpose” in support of a Phase III clinical endpoint. Utilize data sets generated in the EFGH study to effectively bridge between culture and qPCR methods. Potentially conduct a “Type C” meeting with the FDA (or similar regulatory body) to present validation and bridging data sets and gain early input.
**Convene a WHO Policy Workshop**
Convening of a WHO Policy Workshop/LMIC stakeholder meeting to discuss how outcomes from a phase III *Shigella* vaccine study (such as reductions in stunting/growth faltering and AMR/antimicrobial usage) can further support vaccine implementation and uptake in LMICs. Publish recommendations and learnings from the WHO Policy Workshop.
**Revise WHO Preferred Product Characteristics**
Address any recommended modifications to the WHO PPCs based on inputs from the Regulatory Science Workshop and the WHO Policy Workshop. Present modified PPCs to the PDVAC for additional input and recommendations.
**Explore Potential *Shigella* Combination Vaccines**
Develop a strategic framework to identify and structure considerations for combination vaccines, and once developed, apply the framework to vaccines in the current pipeline. In parallel, develop combination vaccine value assessments. Consult extensively with immunization stakeholders and programs at the national and regional levels, including civil society organizations, on potential public health benefits, priority of, and demand for the provisionally preferred combinations identified. Consult with vaccine manufacturers and developers to assess the probability of technical and regulatory success for. Where appropriate, signal potential combinations that could be of interest through WHO PPCs.

## Data Availability

The data presented in this study are available in this article.

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
