# Peer review of "WHO Workshop Report: Regulatory Science to Inform Clinical Pathways for Shigella Vaccines Intended for Use in Children in Low- and Middle-Income Countries"

_vaccines, 2025, doi:10.3390/vaccines13050439_

Round 1

Reviewer 1 Report

Comments and Suggestions for Authors

This is a good manuscript worth of publication. I do have a few observations

1) How would recommend providing access to the vaccine to people who live in poorer areas of the targeted countries? As, you may know in some countries services are provided in the urban areas.

2) How would you recommend making the vaccines accessible and affordable to targeted nations? 

3) How could you utilize NGOs in the process? As you know in some poorer areas they provide the only healthcare instructure?

4) What type of incentives would you recommend for pharmaceutical companies to produce more bivalent or trivalent vaccines targeting pressing world health issues, in particular in children?

Author Response

RESPONSE COMMENTS 1-4:  The authors appreciate the reviewer’s observations, comments, and access-related questions.  These are certainly highly relevant for a Shigella vaccine. However we consider these beyond the scope of this manuscript since they were not discussed as part of our workshop which  focused on key parameters of a phase III vaccine efficacy study.  

While some work has begun on financing and access considerations for Shigella vaccines, these discussions are early and ongoing with country stakeholders and other relevant partners, such as Gavi.  As vaccines move closer to market authorization and WHO pre-qualification, WHO will convene around the topics raised by the reviewer and information from those meetings will be disseminated.

Reviewer 2 Report

Comments and Suggestions for Authors

Estimated Authors,

Estimated Editors,

in this concise and well written paper, Authors provide a detailed summary of the Regulatory Science Workshop held in Nairobi in 2024 on the upcoming Shigella vaccines and corresponding regulatory studies. Briefly, the paper provides to the readers a wholesome description of the characteristics for upcoming studies aimed to define the efficacy of Shigella vaccines, whose characteristics have been otherwise provided across the main text.

The present paper has several strengths, including an accurate description of the content of the meeting, and the well-documented description of the scientific background leading to the choices extensively described and depicted within the main text.

From the point of view of the present reviewers, only minor amendments could be envisaged, and more precisely:

1) The main title could (and should) mirror more precisely the association with the regulatory meeting from Nairobi, for example: "Shigella vaccines intended for use in children in low- and middle-income countries: report from a WHO Regulatory Science Workshop (Nairobi, Kenia, 2024) and evidence-based background";

2) Authors should expand their final sections taking into account the potential impact on the original design of current studies of the ongoing retirement of the United States from WHO in terms of funding of vaccination programs. A very short comment would be sufficient for making the paper up-to-date with the current landscape.

Author Response

RESPONSE to COMMENT 1:  The authors appreciate the reviewer’s comments and suggestion for a manuscript title modification.  Although the Regulatory Science Workshop was held in Nairobi, Kenya, a location identified in the text of the manuscript, the outcomes and recommendations from the workshop are applicable to many regions in which a Shigella vaccine phase III efficacy study could be conducted.  Therefore, to reduce any inference that the information disseminated by the paper is specific to a particular region, the authors respectfully request that the title not include the location of the workshop.  

RESPONSE to COMMENT 2:  The reviewer’s comment is well received and certainly pertinent.  Although there will be significant impacts of the US administration’s recent decision to withdraw from WHO, the specific impact on the clinical trial design and the likelihood of conducting  a phase III efficacy study for a Shigella vaccine may not be impacted,  since WHO is not a funding agency.   Impacts on the broader aspects of US withdrawal from WHO and Gavi on vaccination programmes are largely outside the scope of this manuscript.